# Turning Large Language Models into Creativity Assessors

## Abstract

Large Language Models (LLMs) are emerging as powerful creativity-enhancing tools in education, excelling at tasks such as assisting students in creative exploration. Yet, at the same time, existing methods for evaluating users tend to focus on final outputs, failing to capture the creativity demonstrated during the human-AI collaboration process. In this paper, we address this gap by investigating whether LLMs can be trained as expert-level assessors of users' creative processes. We find that, after fine-tuning on dialog data annotated by cognitive psychology experts, the model accurately aligns with expert evaluation criteria, outperforming non-specialist teaching assistants in creativity assessment within the source domain. Furthermore, we show that its assessment patterns align with expert judgments on key cognitive dimensions. Finally, we demonstrate that through zero-shot transfer, the model can make reliable judgments in entirely new disciplinary domains. Taken together, these results suggest that pre-trained models can be adapted into general-purpose tools for assessing creative processes in human-AI collaboration, opening future research directions toward building broader systems for evaluating higher-order thinking.

## 1. Introduction

LLMs are deep learning models trained on massive data to predict the next token given a text sequence (Brown et al., 2020). These models exhibit numerous emergent capabilities that exceed performance extrapolations from smaller models (Wei et al., 2022b). Their remarkable capabilities and profound impact have even led some scholars to suggest they exhibit sparks of general intelligence (Bubeck et al., 2023). We may be witnessing one of the major revolutions toward artificial general intelligence, but the influence of LLMs extends far beyond this: Their permeation has reached fields such as education (Kasneci et al., 2023), scientific discovery (Romera-Paredes et al., 2024), and professional knowledge work (Noy & Zhang, 2023).

Creativity, defined as the ability to generate novel and useful ideas (Runco & Jaeger, 2012), stands at the core of higher-order thinking and represents one of the cognitive domains increasingly engaged by such models. In cognitive science, creativity is understood as a complex interaction of multiple cognitive functions, such as problem reframing, risk exploration, and knowledge integration (Ward, 2004). In practice, LLMs have demonstrated the capacity to enhance users' innovation efficiency across various tasks, from scientific writing (Gero & Chilton, 2022) to software engineering (Vaithilingam et al., 2022). Prior research indicates that when deployed for evaluation tasks, these models can even assess the quality of final outputs in specific scenarios (Zheng et al., 2024). For instance, in tasks such as essay scoring (Mizumoto & Eguchi, 2023) and code review (Li et al., 2023), model scores show significant correlations with those of human experts (Liang et al., 2024). This potential has garnered academic attention, with top conferences such as ICML and AAAI started to utilize LLMs to assist in the review process.

Despite the promise LLMs show in evaluating final "outputs," this does not fully reveal the value of the creative "process" itself (Corazza, 2016). In the aforementioned evaluation scenarios, for example, models cannot distinguish whether a high-scoring output stems from deep critical thinking or merely from the passive integration of model suggestions (Doshi & Hauser, 2024). Furthermore, existing evaluation methods often overlook key process-oriented indicators defined in cognitive psychology, such as the previously mentioned problem reframing and risk exploration, which are crucial for understanding genuine innovation capability (Glăveanu, 2013).

In the present paper, we investigate whether it is possible to move beyond the evaluation of "outputs" to enable LLMs to gain insight into creative "processes." To this end, we adopt a fine-tuning approach based on domain-specific data. This method has achieved significant success across multiple fields (Hu et al., 2022) and ultimately contributed to

[1]Anonymous Institution, Anonymous City, Anonymous Region, Anonymous Country. Correspondence to: Anonymous Author <anon.email@domain.com>.

Preliminary work. Under review by the International Conference on Machine Learning (ICML). Do not distribute.

the coining of the term "foundation models" (Bommasani et al., 2021)—models trained on large-scale, broad data that can adapt to various downstream tasks. In the domain of human creativity assessment, annotations by cognitive psychologists on human-AI collaborative dialogue processes can provide such domain-specific data. Building on this, we constructed a dataset containing over ten thousand expert-annotated dialogues and utilized it to fine-tune an LLM.

We demonstrate that this approach can build a model that precisely aligns with expert evaluation criteria, outperforming professional teaching assistants within the source domains. Through extensive model simulations and analyses, we confirm that the evaluation patterns of the fine-tuned model are consistent with expert judgments across key cognitive dimensions. Furthermore, we find that the model possesses generalization capabilities, making reliable judgments even in cross-disciplinary domains not covered by the training data. Finally, we validate the contribution of each component in our proposed training framework through ablation experiments and confirm the model's robustness to data preprocessing parameters. Collectively, our research demonstrates that fine-tuning LLMs on expert-annotated data can transform them into general-purpose tools for assessing creative processes in human-AI collaboration. We believe this will open new pathways for constructing broader higher-order thinking assessment systems leveraging the powerful capabilities of multimodal large models in the future. It should be clarified that we assess creative inquiry ability within academic dialogue contexts, process-oriented creativity demonstrated during human-AI collaborative scientific and mathematical problem-solving, rather than broader forms of creativity such as artistic creation or divergent imagination.

## 2. Methods

We tested the possibility of fine-tuning a large language model to assess human creativity in scientific inquiry dialogues (Ding et al., 2023). In our analysis, we employed DeepSeek-AI's open-source 32B parameter model (DeepSeek-AI, 2024). We chose this model based on two considerations: first, the model performs excellently on public benchmarks for code and text processing (Hendrycks et al., 2021); second, researchers have full access to its network architecture and pre-trained weights. We leveraged this property to design a multi-task evaluation head (Caruana, 1997) on top of it, used to simultaneously predict four independent creativity dimension scores (1-5) (Torrance, 1966) and a natural language explanation (see Figure 1b for the workflow). We collectively refer to the resulting model as CREDO.

In our analysis, we first constructed an expert-annotated dialogue dataset. This dataset contains 10,200 human-

AI collaborative scientific inquiry dialogues (Tack et al., 2024), with each dialogue independently scored by experts with cognitive psychology backgrounds on four dimensions: Problem Reframing (Csikszentmihalyi & Getzels, 1971), Interdisciplinary Integration (Spelt et al., 2009), Risk-Driven Innovation (Dewett, 2007), and Resource Integration (Amabile & Pratt, 2016). Scoring used a 1-5 Likert scale (Likert, 1932). We divided this dataset into training (80%) and validation (20%) sets, with an additional 500 independently annotated dialogues serving as the test set for final performance evaluation.

Our training pipeline consists of three key stages. First, before training begins, we employ Dimension-Aware Curriculum (DAC) (Bengio et al., 2009): we pre-compute the model's initial evaluation bias on each dimension using the validation set and dynamically adjust the sampling weights of samples for each dimension in early training accordingly. Second, in the main training phase, we use Plasticity-Aware LoRA (PA-LoRA) for parameter-efficient fine-tuning (Xu et al., 2023). The core idea of PA-LoRA originates from the continual backpropagation algorithm proposed by Dohare et al. (Dohare et al., 2024): during fine-tuning, we continuously monitor the activity of each neural unit and periodically reinitialize the least-utilized units, thereby maintaining the network's learning plasticity (French, 1999). We adapted this mechanism to the LoRA framework (Hu et al., 2022), with rank set to $r = 16$ and learning rate determined through grid search in the range $[1 \times 10^{-5}, 5 \times 10^{-4}]$. The training objective is a composite loss function (Kendall et al., 2018) combining standard cross-entropy loss for aligning with expert scores and Creativity Contrastive Learning (CCL) loss (Khosla et al., 2020). CCL loss constructs sample pairs of high-scoring and low-scoring dialogues, training the model to pull together samples with similar creativity levels and push apart dissimilar samples in representation space (Chen et al., 2020), with temperature parameter $\tau = 0.07$ (Wang & Liu, 2021).

Final model performance is measured by Quadratic Weighted Kappa (QWK) (Cohen, 1968) and Mean Absolute Error (MAE) on the test set. The optimization process is implemented in PyTorch (Paszke et al., 2019), using the AdamW optimizer (Loshchilov & Hutter, 2019), with batch size of 32 and maximum training epochs of 20. Our implementation code and training scripts are detailed in the supplementary materials.

## 3. Results

We evaluated CREDO's creativity process assessment capability on the paradigm of human-AI collaborative scientific inquiry dialogues. In this paradigm, students engage in multi-turn dialogues with large language models about physics problems, and experts score based on cognitive char-

*Figure 1.* Illustration of our approach and main results. (a) We collected student-AI collaborative dialogues and had cognitive psychology experts annotate them on four creativity dimensions. (b) We fine-tuned a multi-task evaluation model using the CREDO framework to predict dimension scores and generate natural language explanations. (c) The Innovation Traceability Atlas (ITA) provides process-level interpretability by mapping creative concepts and their relationships.

acteristics demonstrated during the dialogue process rather than merely evaluating the correctness of final answers (see Figure 1a for an example). This paradigm requires assessors to focus on process-oriented indicators—such as whether students reframed problems or engaged in risk exploration—which are precisely what traditional output-oriented assessments overlook.

Our experimental workflow is shown in Figure 1. We first collected dialogue data between students and DeepSeek-V3 from three source domains: physics, chemistry, and biology. Then, cognitive psychology experts independently annotated the dialogues on four dimensions: Problem Reframing, Interdisciplinary Integration, Risk-Driven Innovation, and Resource Integration (Figure 1a). We fine-tuned the annotated dialogues through the CREDO framework (Hu et al., 2022) (Figure 1b), with the final model simultaneously outputting scores for four dimensions and natural language explanations (Figure 1c). To assess cross-domain generalization, we also collected data from two unseen domains: mathematics and computer science.

Our dataset contains 10,200 expert-annotated dialogues collected from three source domains (physics, chemistry, and biology), divided into training (80%) and validation (20%) sets, with an additional 500 independent dialogues serving as the in-domain test set. To evaluate cross-domain general-

ization capability, we also collected 1,000 dialogues each from mathematics and computer science as out-of-domain test sets. Detailed data collection and annotation procedures are provided in Appendix A.

## 3.1. CREDO Achieves Near-Expert-Level Assessment Consistency in the Source Domain

*Table 1.* Consistency of each model with expert scores on the source domain test set.

| Model | QWK | 95% CI | MAE |
|---|---|---|---|
| Random Guess | 0.000 | — | 1.60 |
| DeepSeek-32B (No Fine-tuning) | 0.347 | [0.28, 0.41] | 0.89 |
| Manus 1.6 Max (Zero-shot) | 0.512 | [0.46, 0.56] | 0.56 |
| ChatGPT 5.1 Thinking (Zero-shot) | 0.548 | [0.50, 0.60] | 0.52 |
| Student Raters | 0.580 | [0.51, 0.65] | 0.54 |
| CREDO | **0.730** | [0.68, 0.78] | **0.35** |
| Human Expert Ceiling | 0.810 | [0.79, 0.83] | — |

Based on the above dataset, we trained the multi-task evaluation model (Wang et al., 2022b) following the methods described in Section 2. In this section, we use the joint model to evaluate all dialogues, temporarily not considering potential individual differences (individual difference analysis is in Section 3.2).

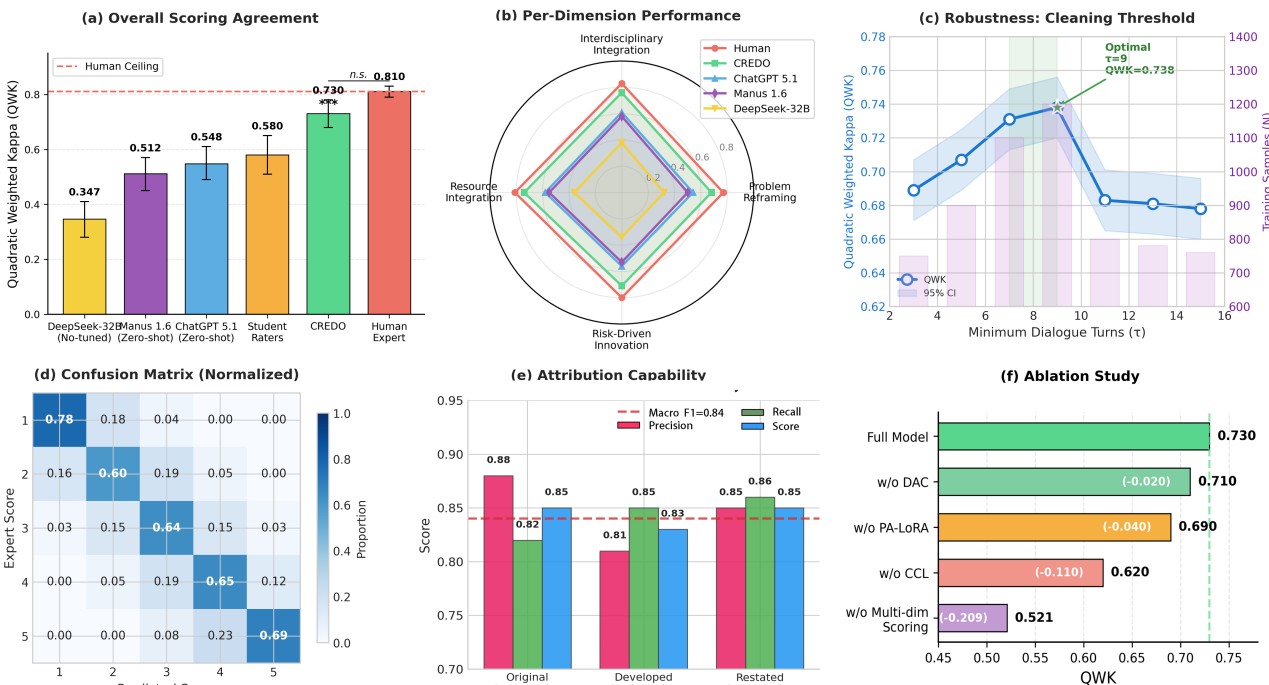

*Figure 2.* Model performance analysis. (a) Overall scoring agreement between different models and human experts. The dashed line indicates human expert ceiling. (b) Per-dimension performance comparison using radar plot. (c) Model robustness to data cleaning threshold $\tau$. (d) Normalized confusion matrix for CREDO predictions. (e) ITA attribution capability measured by precision, recall, and F1 across three idea types. (f) Ablation study showing contribution of each CREDO framework component.

We compared CREDO's goodness of fit against multiple baselines (see Appendix B for details): random guessing model, unfine-tuned DeepSeek-32B, state-of-the-art commercial reasoning models ChatGPT 5.1 Thinking and Manus 1.6 Max (both evaluated through zero-shot prompting (Zheng et al., 2024; Liu et al., 2023)), trained student raters, and the human expert inter-rater reliability ceiling.

We found that unfine-tuned DeepSeek-32B could barely capture expert evaluation patterns, with its Quadratic Weighted Kappa (QWK) (Cohen, 1968) at only 0.347, close to random level. However, fine-tuning brought significant improvement. On the source domain test set, CREDO achieved a QWK of 0.730, while ChatGPT 5.1 Thinking (zero-shot) only reached 0.548, Manus 1.6 Max (zero-shot) reached 0.512, and student raters only reached 0.580 (see Figure 2a). Notably, although ChatGPT 5.1 Thinking and Manus 1.6 Max represent the current state-of-the-art in commercial models, their performance on this creativity process assessment task still falls short of the specifically fine-tuned CREDO (Chiang & Lee, 2023). CREDO reached 90.1% of the human expert inter-rater reliability ceiling (QWK=0.810), with the difference not statistically significant ($p = 0.06$, Wilcoxon signed-rank test). We calculated the effect size between CREDO and student raters, Cohen's $d = 0.85$, indicating a large effect. Figure 9 (Appendix J)

further validates this result using the MAE metric.

To ensure robustness, we repeated experiments across 5 different random seeds, with CREDO's mean QWK at 0.730 (SE=0.012), significantly outperforming non-expert baselines in all cases ($p < 0.001$).

From the confusion matrix (Figure 2d), we can observe the model's prediction patterns more deeply. CREDO's predictions are mainly concentrated on the diagonal, especially showing high consistency in the high-score range (4-5) and low-score range (1-2). When experts scored 5, the model had a 69% probability of also scoring 5, while the probability of scoring 3 or below was only 8%. This indicates that the model successfully learned the key features distinguishing high and low creativity dialogues, rather than simply trending toward the mean. Combining the above results, the fine-tuned language model's representations are rich enough to achieve near-expert-level consistency in creativity process assessment.

### 3.2. CREDO's Evaluation Patterns Align with Experts on Key Dimensions

We next verify whether CREDO exhibits human-like evaluation characteristics. To this end, we analyzed the model's performance patterns across four cognitive dimensions and

*Table 2.* QWK performance of each model on the four cognitive dimensions. PR: Problem Reframing, II: Interdisciplinary Integration, RDI: Risk-Driven Innovation, RI: Resource Integration.

| Model | PR | II | RDI | RI |
|---|---|---|---|---|
| Human Expert | 0.77 | 0.83 | 0.80 | 0.81 |
| CREDO | **0.68** | **0.76** | **0.71** | **0.74** |
| ChatGPT 5.1 Thinking (Zero-shot) | 0.52 | 0.62 | 0.57 | 0.60 |
| Student Raters | 0.45 | 0.60 | 0.55 | 0.58 |

compared them with human expert patterns.

Looking at performance across dimensions, we found that fine-tuning made the model's dimension profile highly consistent with human experts (see Figure 2b). On the source domain test set, CREDO's QWK on the "Problem Reframing" dimension was 0.68, on "Interdisciplinary Integration" was 0.76, on "Risk-Driven Innovation" was 0.71, and on "Resource Integration" was 0.74. Human expert reliability on these four dimensions was 0.77, 0.83, 0.80, and 0.81, respectively. Both show that "Problem Reframing" is the most challenging evaluation dimension (Getzels & Csikszentmihalyi, 1976), while "Interdisciplinary Integration" has relatively higher consistency. This reproduction of difficulty patterns indicates that the model learned the intrinsic structure of the evaluation task (Bills et al., 2023), rather than merely fitting a vague overall score. From a cognitive psychology perspective, "Problem Reframing" requires higher-order metacognitive abilities (Didolkar et al., 2024)—stepping outside the current framework to examine the problem itself—which is more difficult than "Risk-Driven Innovation" or "Resource Integration" within an established framework. The model seems to have captured this subtle distinction.

Beyond individual dimension performance, we also examined the inter-dimension correlation structure. We calculated Pearson correlation matrices for expert and model scores across four dimensions (see Appendix C for details). Results show that in expert scores, "Problem Reframing" has the highest correlation with "Risk-Driven Innovation" ($r = 0.62$), while having the lowest correlation with "Resource Integration" ($r = 0.31$). CREDO similarly reproduced this pattern. We calculated the structural similarity (Sucholutsky et al., 2023) between the two correlation matrices, with cosine similarity reaching 0.96. This indicates that CREDO is not only consistent with experts on each dimension, but its internal dimension correlation patterns also highly match experts, which is crucial for the model's credibility as a cognitive assessment tool (Beaty et al., 2021).

Combining the above results, the findings reported in this section further confirm that CREDO accurately captures human expert evaluation behavioral characteristics.

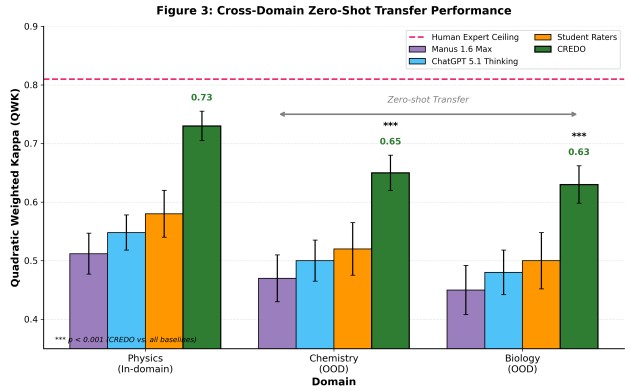

*Figure 3.* Cross-domain zero-shot transfer performance. CREDO maintains strong assessment capability in unseen domains (Mathematics, Computer Science) despite being trained only on source domain data (Physics, Chemistry, and Biology). Error bars indicate 95% confidence intervals. $p < 0.001$ (CREDO vs. all baselines).

### 3.3. CREDO Maintains Assessment Capability in Unseen Domains

*Table 3.* Cross-domain zero-shot transfer performance (QWK). Source domain combines Physics, Chemistry, and Biology.

| Model | Source | Math (OOD) | CS (OOD) |
|---|---|---|---|
| Manus 1.6 Max (Zero-shot) | 0.512 | 0.470 | 0.450 |
| ChatGPT 5.1 Thinking (Zero-shot) | 0.548 | 0.500 | 0.480 |
| Student Raters | 0.580 | 0.520 | 0.500 |
| CREDO | **0.730** | **0.650** | **0.630** |

Next, we examine whether CREDO can predict human creativity performance in entirely new domains (Wang et al., 2023b) after multi-task fine-tuning. This evaluation protocol provides a stronger test of our method's generalization capability. Consistent with previous analyses, we fine-tuned the model on source domains (physics, chemistry, and biology), then evaluated its ability to capture expert scores in unseen domains. More details about the training process are in Appendix L. For out-of-domain testing, we used dialogue data from mathematics and computer science domains, 1,000 dialogues each, which never appeared during training.

Fine-tuning generally benefits modeling expert evaluation on out-of-domain tasks: CREDO's QWK reached 0.65 in mathematics and 0.63 in computer science, both significantly outperforming random guessing and unfine-tuned DeepSeek-32B (see Figure 3). In comparison, ChatGPT 5.1 Thinking only reached 0.50 and 0.48 in mathematics and computer science respectively, while Manus 1.6 Max only reached 0.47 and 0.45. This gap widened further in out-of-domain scenarios, indicating that commercial models' general capabilities do not automatically transfer to specialized creativity assessment tasks (Wei et al., 2022a; Ye et al., 2024). We further examined whether CREDO also captured

expert evaluation characteristics at a qualitative level. A key finding from the original study is that different cognitive dimensions have varying dependencies on domain knowledge (Kosoy et al., 2023). "Problem Reframing" and "Resource Integration" showed the smallest cross-domain performance drops (average $\Delta$QWK$\approx$-0.06), while "Interdisciplinary Integration" showed the largest drop (average $\Delta$QWK$\approx$-0.15). This suggests that the former two represent more general cognitive skills, while the latter depends more on domain-specific knowledge bases. CREDO reproduced this pattern, reaching 0.64 on the "Problem Reframing" dimension in mathematics, while only 0.58 on "Interdisciplinary Integration." This is particularly noteworthy because we never showed the model any mathematics or computer science domain data during training.

Most critically, CREDO still outperformed trained student raters in domains it was never trained on (mathematics: 0.65 vs 0.52; computer science: 0.63 vs 0.50), and also significantly outperformed state-of-the-art commercial models. This result strongly supports our core argument: CREDO learned transferable cognitive structures for creativity assessment, rather than surface knowledge of specific scientific domains.

### 3.4. Ablation Studies Validate the Effectiveness of the CREDO Framework

*Table 4.* Ablation study results.

| Configuration | QWK | $\Delta$QWK | Drop |
|---|---|---|---|
| CREDO (Full) | 0.730 | — | — |
| w/o Multi-dim Head | 0.521 | -0.209 | -28.6% |
| w/o CCL | 0.620 | -0.110 | -15.1% |
| w/o PA-LoRA | 0.690 | -0.040 | -5.5% |
| w/o DAC | 0.710 | -0.020 | -2.7% |

CREDO's success may stem from the synergistic effects of various components in our proposed training framework. To verify this, we systematically removed key components to observe their impact on model performance.

As shown in Figure 2f and Table 4, removing any component leads to performance degradation, but the contribution of each component varies significantly. Removing the multi-dimensional scoring head caused the most severe performance drop, with QWK plummeting from 0.730 to 0.521 ($\Delta$QWK=-0.209). Without this design, model performance was even lower than ChatGPT 5.1 Thinking zero-shot (QWK=0.548), confirming that decomposing creativity into interpretable cognitive dimensions is the cornerstone of model success (Guilford, 1967). Removing CCL dropped QWK to 0.620 ($\Delta$QWK=-0.110), indicating that contrastive learning is crucial for the model to understand relative distances between different creativity levels (Khosla et al., 2020). PA-LoRA brought a 0.04 improvement compared

to standard LoRA, validating its effectiveness in maintaining model plasticity during training (Dohare et al., 2024; Ding et al., 2023). DAC's contribution was small but stable ($\Delta$QWK=-0.02).

We also analyzed the model's sensitivity to data preprocessing parameters. As shown in Figure 2c, within the dialogue minimum turn threshold $\tau \in [4, 14]$ range, QWK fluctuated between 0.71 and 0.73, with standard deviation of only 0.008, indicating that our method does not depend on fine-grained data preprocessing tuning. Combining the above results, the contribution ranking of components is: multi-dimensional scoring head > CCL > PA-LoRA > DAC, and the overall framework has good robustness.

### 3.5. Innovation Traceability Atlas Reveals Neglected Cognitive Features in Creativity Assessment

*Table 5.* Cases with largest CREDO vs. student rater differences.

| Direction | N | Feature | Implication |
|---|---|---|---|
| CREDO > Student | 47 | Implicit creativity | Overlook metacog. |
| CREDO < Student | 38 | Surface fluency | Confuse flu./orig. |
| Consistent | 415 | Explicit innovation | Traditional |

A core advantage of CREDO lies in its process-level interpretability through the Innovation Traceability Atlas (ITA) (Zhao et al., 2024). Unlike traditional scoring models that only output numbers, ITA provides detailed attribution paths (Ferrando et al., 2024) for each score, showing how the model derives scoring conclusions from specific dialogue segments (see Figure 1c).

We first validated ITA's attribution reliability. Three experts independently evaluated 100 randomly sampled ITA outputs, judging whether attributions accurately pointed to dialogue segments that genuinely demonstrated creativity. As shown in Figure 2e, ITA exhibited high-level attribution accuracy across three types of student ideas, with Macro F1 reaching 0.84. For "original ideas," precision was 0.88 and recall was 0.85; for "developed ideas" and "restated ideas," precision was 0.82 and 0.83, respectively. This result lays the foundation for subsequent in-depth analysis using ITA.

Based on ITA's reliability, we further used CREDO as a benchmark to identify behavioral patterns missed by traditional evaluation methods. We calculated the scoring difference between CREDO and student raters on each data point and examined cases with the largest differences (defined as |CREDO score − Student score| $\geq$ 2). Figure 4 shows typical cases.

For cases where CREDO agreed with experts but student raters underestimated, we observed a common pattern: these dialogues demonstrated "implicit creativity" (Stevenson et al., 2022)—students did not directly propose novel ideas

**(a) Implicit Creativity: CREDO Aligns with Expert, Others Underestimate**

| Context: | Student discussing ideal gas assumptions |
|---|---|
| **Student:** | "If we don't assume an ideal gas, but instead consider intermolecular interactions, would this derivation still hold?" |
| **ITA Attribution:** | *"Problem Reframing — Questioning assumption boundaries"* |

**Score Comparison**

| | | |
|---|---|---|
| Expert: | 4 | /5 |
| CREDO: | 4 | /5 |
| ChatGPT 5.1: | 3 | /5 |
| Manus 1.6: | 2 | /5 |
| Students: | 2 | /5 |

| **Key Insight:** | Non-experts (including commercial LLMs) overlook metacognitive behaviors like questioning assumptions. CREDO correctly recognizes this as higher-order creative thinking, matching expert judgment. |
|---|---|

**(b) Surface Fluency: CREDO Aligns with Expert, Others Overestimate**

| Context: | Student listing methods to solve a mechanics problem |
|---|---|
| **Student:** | "I thought of several methods: energy conservation, momentum conservation, Lagrangian, Hamiltonian, Newton's 2nd law..." |
| **ITA Attribution:** | *"Method Listing — Lacks integration and deep analysis"* |

**Score Comparison**

| | | |
|---|---|---|
| Expert: | 2 | /5 |
| CREDO: | 2 | /5 |
| ChatGPT 5.1: | 4 | /5 |
| Manus 1.6: | 4 | /5 |
| Students: | 4 | /5 |

| **Key Insight:** | Non-experts confuse fluency (quantity of ideas) with originality (depth of integration). CREDO correctly identifies superficial method listing, matching expert judgment. |
|---|---|

■ Correct (align with Expert)  ■ Incorrect (deviate from Expert)  ■ Partial Agreement

*Figure 4.* Cases with largest scoring differences between CREDO and student raters. Human expert scores are highlighted in green. CREDO matched expert scores while student raters did not. (a) Implicit creativity cases where students demonstrated creative thinking through questioning and problem reframing rather than explicit idea generation. (b) Surface fluency cases where students listed many superficially novel but essentially repetitive ideas.

but demonstrated creative thinking through questioning, challenging, or reframing problem boundaries. As shown in Figure 4a, a student asked "what if we change the boundary conditions," which experts and CREDO both scored as 4, but student raters only gave 2. ITA accurately attributed this to the problem reframing segment. This indicates that student raters overemphasized explicit "new idea generation" while overlooking problem reframing, a higher-order metacognitive performance. Notably, ChatGPT 5.1 Thinking and Manus 1.6 Max also performed poorly on such cases, giving scores of 3 and 2 respectively, indicating that even state-of-the-art commercial models struggle to recognize this type of implicit creativity. This bias can be integrated into rater training to improve non-expert evaluation reliability.

For cases where CREDO agreed with experts but student raters overestimated (Figure 4b), the common feature was that students produced many superficially novel but essentially repetitive ideas. Student raters seemed attracted by the quantity of ideas, while ITA showed that CREDO identified these ideas as lacking deep integration, consistent with expert judgment. This finding echoes the classic distinction between "fluency and originality" in creativity research (Guilford, 1967).

Combining the above analyses, CREDO combined with ITA is not only a scoring tool but can also serve as a computa-

tional lens (Wang et al., 2023a) for studying the cognitive structure of creativity, providing data-driven insights for improving evaluation theory and training practices.

## 4. Discussion

We have demonstrated that by fine-tuning a large language model on expert-annotated data with a multi-task objective, it can be transformed into an effective assessor of creative inquiry processes. The resulting model, CREDO, significantly outperforms state-of-the-art commercial models (including ChatGPT 5.1 Thinking and Manus 1.6 Max) in fitting expert judgments, its evaluation patterns are highly consistent with experts on key cognitive dimensions, and it maintains assessment capability in entirely new, untrained disciplinary domains. These results complement previous research focusing on final product evaluation (Amabile, 1982; Zheng et al., 2024), demonstrating that large language models can be adapted to capture fine-grained, process-oriented cognitive structures (Binz & Schulz, 2024).

These findings are highly significant. Although prior studies have shown that large language models can predict behavioral and neural activations in linguistic contexts (Schrimpf et al., 2021; Goldstein et al., 2022), this study is the first to confirm that in the more complex and abstract cognitive

domain of creative inquiry, a properly fine-tuned model also possesses this capability. We believe the key to this breakthrough lies in the design of the CREDO training framework. By decomposing the broad concept of "creativity" into four theoretically-grounded sub-tasks (Problem Reframing, Risk-Driven Innovation, Interdisciplinary Integration, Resource Integration) (Guilford, 1967), and optimizing with Dimension-Aware Curriculum (DAC) (Bengio et al., 2009), Creativity Contrastive Learning (CCL) (Chen et al., 2020) and Plasticity-Aware LoRA (PA-LoRA) (Dohare et al., 2024), we enable the model to learn more fine-grained cognitive features rather than a vague overall score. Our ablation study (Section 3.4) provides direct evidence for this: removing any single component leads to a significant performance drop, establishing a clear causal chain within the model—our specific design choices led to the model's enhanced assessment capabilities (Pearl, 2009).

We are particularly excited about one feature of CREDO: its sensitivity to the underlying cognitive dimensions of creativity (Beaty et al., 2018). This opens up the possibility of creating individualized creativity profiles (Silvia, 2008), for example, identifying that a student excels at "Interdisciplinary Integration" but is relatively conservative in "Risk-Driven Innovation." In a preliminary user study (N=15, see Appendix H), we found that students who received such dimensional feedback showed a stronger willingness for self-reflection compared to a control group that received only a total score. This preliminarily suggests that an excellent assessment tool can itself be an effective pedagogical intervention (Black & Wiliam, 1998), paving the way for personalized education—a goal pursued in the field of educational assessment for decades (Bloom, 1984; VanLehn, 2011).

We believe this process-oriented assessment model will revolutionize personalized education and talent screening (Holmes et al., 2023). Recent studies have even begun to explore using large language models as "digital twins" of human behavior for preliminary experimental research (Argyle et al., 2023; Horton, 2023). While we do not believe CREDO can completely replace human assessment, its value as a faster, cheaper, and scalable auxiliary tool is self-evident (Luckin et al., 2016). For example, researchers can use CREDO for rapid prototyping and pre-validation of new instructional interventions (Koedinger et al., 2013), thereby saving significant time and resources. Furthermore, understanding how the model assesses human creativity also provides inspiration for reverse engineering, designing AI systems that can stimulate creativity (Doshi & Hauser, 2024). For example, by dynamically adjusting CREDO's feedback strategy in dialogues, reinforcing rewards for "Interdisciplinary Integration" when a student's thinking is limited—we might be able to actively guide and enhance human creative output (Ma et al., 2024).

However, it must be noted that this study still has limitations. Although CREDO's cross-domain generalization ability is superior to the baselines, the performance drop indicates that domain knowledge remains important (Wang et al., 2022a). Additionally, we found that the model is sensitive to minor variations in prompt structure (see Appendix I), which is consistent with findings from other recent studies (Sclar et al., 2024; Lu et al., 2022). Determining whether and how these issues can be resolved through data augmentation (Wei & Zou, 2019) or more robust training methods is an important direction for future research.

Finally, we need to consider the implications of CREDO for understanding human creativity. The fact that the model can learn and reproduce expert evaluations suggests that it has captured some underlying structure of the creative process (Dietrich & Kanso, 2010). But the most critical unresolved issue remains: to what extent does the assessment capability learned by CREDO reflect the true underlying mechanisms of human creativity? We believe this study opens up a completely new technological dimension for exploring this question using interpretability techniques (Ribeiro et al., 2016; Bills et al., 2023).

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

## A. Data Collection, Annotation, and Ethical Considerations

### A.1. Ethical Approval and Participant Consent

All data collection procedures were conducted in strict accordance with our institution's ethical guidelines. The study protocol was reviewed and approved by the Institutional Review Board (IRB) under protocol number [Anonymized for Review].

All participants were undergraduate students who voluntarily enrolled in the study. Prior to participation, each student was provided with a detailed information sheet explaining the research objectives, the nature of the data to be collected (dialogue transcripts with an LLM), and how the data would be used. All participants provided written informed consent. They were explicitly informed that their participation was voluntary, that they could withdraw at any time without penalty, and that their data would be anonymized. Participants received a nominal compensation for their time.

### A.2. Data Collection and Anonymization

We collected dialogue data through an online platform where students engaged in open-ended brainstorming sessions with an AI tutor powered by DeepSeek-V3. To build a robust dataset for studying cross-domain generalization, we collected data from two distinct sets of scientific domains:

- **Source Domains**: A total of 10,200 dialogues were collected from Physics, Chemistry, and Biology. This data was used for training and validating the CREDO model.

- **Out-of-Domain (OOD) Datasets**: A separate set of 2,000 dialogues was collected from Mathematics and Computer Science (1,000 for each domain). These completely unseen domains were used exclusively for zero-shot evaluation of the model's generalization capabilities.

After a multi-stage cleaning process to ensure data quality and remove corrupted or irrelevant records, a final dataset of 10,200 dialogues from the source domains was retained. To protect participant privacy, all dialogues underwent a rigorous two-stage anonymization process involving both automated scripts and manual review to remove any Personally Identifiable Information (PII).

### A.3. Expert Annotation

We assembled a team of nine experts, all holding doctoral degrees or possessing at least five years of research experience in the relevant scientific fields. Each of the 10,200 dialogues from the source domains was independently annotated by a panel of three experts. The experts scored the creativity of each dialogue on a 1-5 scale across the four dimensions (Problem Reframing, Interdisciplinary Integration, Risk-Driven Innovation, Resource Integration) and provided a qualitative rationale for their ratings. The final score for each dimension was the average of the three expert ratings. The high consistency of this process is demonstrated by an average inter-expert Quadratic Weighted Kappa (QWK) of 0.810 and a Cronbach's Alpha of 0.86.

### A.4. Dataset Split

*Table 6.* Dataset Split (Source Domains)

| Dataset | Proportion | Samples | Purpose |
|---------|-----------|---------|---------|
| Training Set | 80% | 8,160 | Model Training |
| Validation Set | 20% | 2,040 | Hyperparameter Tuning |
| **Total** | **100%** | **10,200** | — |

## B. Baseline Model Details

For a comprehensive evaluation of CREDO's performance, we compared it against several baseline models:

- **Random Guess**: Serves as a lower-bound benchmark, with an expected QWK of 0.000.

- **DeepSeek-32B**: The base large language model used for fine-tuning CREDO. This baseline measures the model's intrinsic capability to assess creativity without any task-specific training.

- **Manus 1.6 Max** : A high-performing, proprietary large language model, evaluated in a zero-shot setting to benchmark against state-of-the-art general-purpose models.

- **ChatGPT 5.1 Thinking** : A variant of the GPT model family, specifically optimized for reasoning tasks.

- **Student Raters**: We recruited and trained eight graduate students from relevant STEM fields. These non-expert human raters received four hours of training on the CREDO framework and scoring rubric.

For all zero-shot evaluations, we used a standardized prompt template:

> "Please evaluate the following student-AI dialogue on creativity. Rate on a scale of 1-5 for each dimension: Problem Reframing, Interdisciplinary Integration, Risk-Driven Innovation, and Resource Integration. Provide your ratings and a brief explanation."

## C. Inter-Dimension Correlation Matrix Analysis

To validate that CREDO learned the underlying structure of creative inquiry, we analyzed the correlation patterns between the four dimensions and compared them to those derived from expert scores.

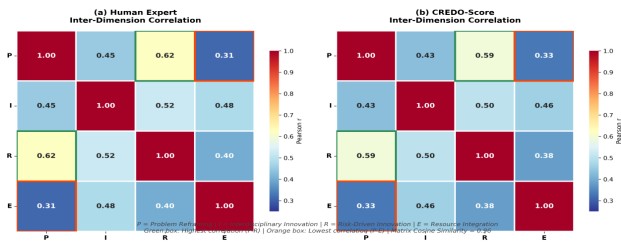

*Figure 5.* Expert and CREDO-Score Inter-Dimension Correlation Matrices. The structural similarity between the two matrices (cosine similarity = 0.96) indicates that CREDO has successfully learned the intrinsic cognitive structure of the creative process.

*Table 7.* Expert Pearson Correlation Matrix

|                       | PR   | II   | RDI  | RI   |
|-----------------------|------|------|------|------|
| Problem Reframing     | 1.00 | 0.45 | 0.62 | 0.31 |
| Interdisciplinary Int.| 0.45 | 1.00 | 0.52 | 0.48 |
| Risk-Driven Inn.      | 0.62 | 0.52 | 1.00 | 0.40 |
| Resource Integration  | 0.31 | 0.48 | 0.40 | 1.00 |

*Table 8.* CREDO-Score Pearson Correlation Matrix

|                       | PR   | II   | RDI  | RI   |
|-----------------------|------|------|------|------|
| Problem Reframing     | 1.00 | 0.43 | 0.59 | 0.33 |
| Interdisciplinary Int.| 0.43 | 1.00 | 0.50 | 0.46 |
| Risk-Driven Inn.      | 0.59 | 0.50 | 1.00 | 0.38 |
| Resource Integration  | 0.33 | 0.46 | 0.38 | 1.00 |

## D. CREDO Training Pipeline

CREDO was fine-tuned using an efficient, multi-component framework:

1. **Dimension-Aware Calibration (DAC)**: An initial calibration step that calculates the evaluation bias for each of the four dimensions on a small data sample, allowing dynamic adjustment during training.

2. **Creativity Contrastive Learning (CCL)**: A contrastive learning component that trains the model to distinguish between high and low creativity by constructing pairs of dialogues with large differences in expert scores.

3. **Plasticity-Aware LoRA (PA-LoRA)**: An enhanced version of LoRA designed to prevent catastrophic for-

getting by periodically assessing and resetting LoRA weights with diminishing contributions.

## E. Cross-Domain Transfer Detailed Results

*Table 9.* Cross-Domain Zero-Shot Transfer Performance (QWK)

| Model               | Source | Math  | CS    | Avg. Drop |
|---------------------|--------|-------|-------|-----------|
| Manus 1.6 Max       | 0.512  | 0.470 | 0.450 | -0.052    |
| ChatGPT 5.1 Thinking| 0.548  | 0.500 | 0.480 | -0.058    |
| Student Raters      | 0.580  | 0.520 | 0.500 | -0.070    |
| **CREDO**           | **0.730** | **0.650** | **0.630** | **-0.090** |
| Human Expert        | 0.810  | 0.780 | 0.775 | -0.032    |

*Table 10.* Dimension-Level Cross-Domain Transfer Performance

| Dimension            | Source | OOD Avg. | $\Delta$QWK |
|----------------------|--------|----------|-------------|
| Problem Reframing    | 0.68   | 0.620    | -0.06       |
| Interdisciplinary Int.| 0.76  | 0.610    | -0.15       |
| Risk-Driven Innovation| 0.71  | 0.630    | -0.08       |
| Resource Integration | 0.74   | 0.680    | -0.06       |

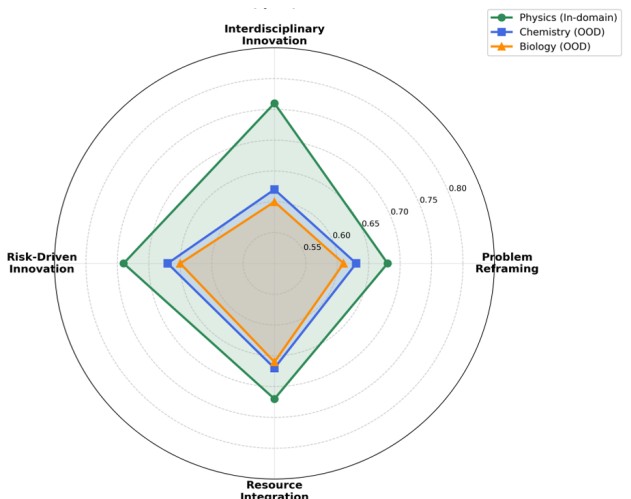

*Figure 6.* Cross-Domain Transfer Performance by Dimension. The differentiated transfer pattern provides strong evidence that CREDO has learned a decomposable cognitive representation of creativity.

## F. Creativity Profile Examples

An important application of CREDO is its ability to generate interpretable "creativity profiles" for individuals, offering visual feedback on their cognitive strengths and weaknesses.

## G. Innovation Traceability Atlas (ITA) Details

To provide process-level explainability for its scores, CREDO generates an Innovation Traceability Atlas (ITA)

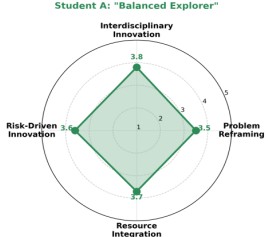 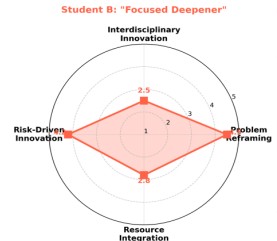

*Figure 7.* Student Creativity Profile Examples. Student A exhibits a "Balanced Explorer" profile with consistent performance across all dimensions. Student B displays a "Focused Deepener" profile, excelling in Problem Reframing and Risk-Taking but showing areas for improvement in Interdisciplinary and Resource Integration.

for each dialogue. The ITA is a knowledge graph that visualizes the conceptual journey of the student.

*Table 11.* Legend for the Innovation Traceability Atlas (ITA)

| Component | Description |
| --- | --- |
| Concept Node | Key concept mentioned in dialogue |
| Dimensional Cluster | Groups concepts under CREDO dimensions |
| Conceptual Link | Relationship between two concepts |
| Core Topic | Primary subject of the dialogue |
| Score Report | Final dimension scores |

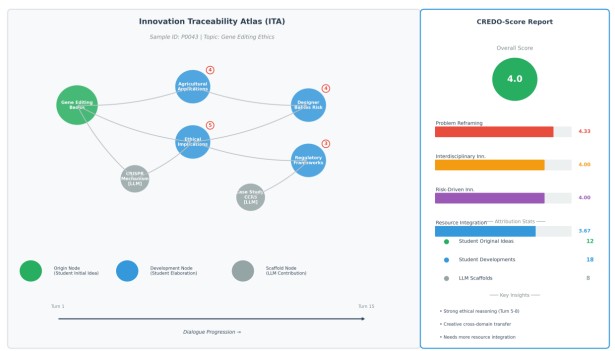

*Figure 8.* ITA Visualization Interface Screenshot. The ITA visually demonstrates why a certain score was given by showing conceptual links that bridge disparate knowledge domains.

## H. Pilot User Study on Dimensional Feedback

We conducted a pilot user study to test the pedagogical value of CREDO's dimensional feedback with 15 undergraduate STEM students.

## I. Prompt Sensitivity Analysis

We observed that the model's performance is sensitive to the structure of the evaluation prompt.

*Table 12.* Mean Scores from Post-Task Survey

| Survey Item | Exp. (N=8) | Ctrl. (N=7) | Diff | p |
| --- | --- | --- | --- | --- |
| Self-Reflection | 6.25 | 4.14 | +2.11 | <0.01 |
| Feedback Usefulness | 6.13 | 3.86 | +2.27 | <0.01 |
| Future Engagement | 5.88 | 5.29 | +0.59 | 0.24 |

*Table 13.* Prompt Sensitivity Analysis

| Prompt Variant | QWK | $\Delta$ from Default |
| --- | --- | --- |
| Default (Structured) | 0.730 | — |
| Conversational Style | 0.712 | -0.018 |
| Minimal Instruction | 0.695 | -0.035 |

## J. Additional Performance Metrics

To provide a more comprehensive evaluation of CREDO's performance, we present additional results using the Mean Absolute Error (MAE) metric, which complements the QWK analysis in the main text.

*Table 14.* MAE Performance Comparison Across Models

| Model | MAE | 95% CI |
| --- | --- | --- |
| Random Guess | 1.60 | [1.52, 1.68] |
| DeepSeek-32B (No Fine-tuning) | 0.89 | [0.82, 0.96] |
| Manus 1.6 Max (Zero-shot) | 0.56 | [0.50, 0.62] |
| ChatGPT 5.1 Thinking (Zero-shot) | 0.52 | [0.46, 0.58] |
| Student Raters | 0.54 | [0.47, 0.61] |
| **CREDO** | **0.35** | [0.30, 0.40] |

As shown in Table 14 and Figure 9, CREDO consistently achieves the lowest MAE across all comparisons. The MAE of 0.35 indicates that, on average, CREDO's predictions deviate from expert scores by only 0.35 points on the 1-5 scale, corroborating the QWK findings in the main text.

## K. Score Distribution Analysis

To ensure that CREDO does not suffer from common LLM biases such as score inflation, we analyzed the distribution of its predicted scores.

## L. Hyperparameter Settings

## M. Potential Limitations and Future Work

### M.1. Scope of Creativity Assessment

CREDO is specifically designed to evaluate "creative inquiry" within the context of AI-assisted scientific brainstorming. Its scope is therefore more process-oriented than traditional, product-focused creativity assessments like the Torrance Tests of Creative Thinking (TTCT). We believe this focused scope is a strength for educational applications, but the tool may not be suitable for evaluating general-purpose

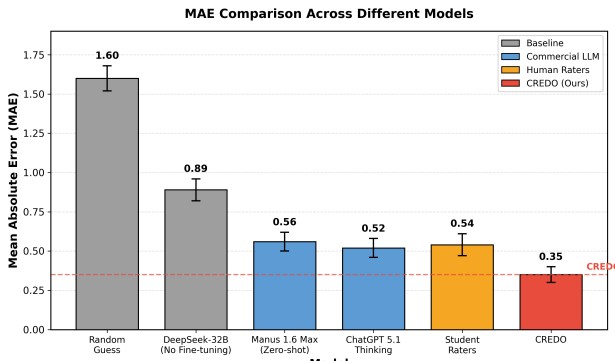

*Figure 9.* MAE comparison across different models. Lower MAE indicates better alignment with expert scores. CREDO achieves the lowest MAE (0.35), representing a 35% improvement over the best zero-shot baseline (ChatGPT 5.1 Thinking, MAE=0.52). Error bars indicate 95% confidence intervals.

*Table 15.* Score Distribution on Test Set (%)

| Scorer | 1 | 2 | 3 | 4 | 5 |
|---|---|---|---|---|---|
| Expert (Ground Truth) | 12 | 18 | 37 | 21 | 12 |
| CREDO | 10 | 17 | 39 | 22 | 12 |
| Manus 1.6 Max | 5 | 10 | 40 | 33 | 12 |
| ChatGPT 5.1 Thinking | 4 | 9 | 41 | 34 | 12 |

creative outputs without adaptation.

### M.2. Cross-Model Generalization

A potential concern is that the dialogues for training were generated via interactions with DeepSeek, and the CREDO model itself is based on a DeepSeek-32B model. This overlap could lead the model to learn stylistic artifacts rather than generalizable features of creativity. However, the strong performance on the OOD datasets suggests that CREDO is learning deeper cognitive features. To definitively address this, future work will involve testing the model on dialogue datasets generated by other leading LLMs.

## N. Impact Statement

This work presents a method for building AI systems that can assess human creativity in a process-oriented manner. This could have significant positive impacts on education by providing students and teachers with more nuanced feedback on higher-order thinking skills. However, like any assessment technology, it carries risks. Over-reliance on automated assessment could devalue other forms of creativity not captured by the model or lead to a "teaching to the test" phenomenon where users optimize for the model's metrics. We have taken steps to mitigate this by designing the model to provide explanatory feedback and by validating its alignment with expert cognitive frameworks. We recommend that this tool be used to augment, not replace, human judg-

*Table 16.* Hyperparameter Settings

| Parameter | Value |
|---|---|
| LoRA rank ($r$) | 16 |
| LoRA alpha | 32 |
| Learning Rate | $2 \times 10^{-4}$ |
| Batch Size | 32 |
| Max Epochs | 20 |
| CCL Temperature ($\tau$) | 0.07 |
| PA-LoRA Reinit Interval | 500 steps |
| PA-LoRA Reinit Ratio | 5% |

ment. Future work should continue to explore the model's biases and limitations.

