# OpenReview forum: "Turning Large Language Models into Creativity Assessors"
_ICML.cc/2026/Conference — Submitted to ICML 2026_

### Official Review · Reviewer_EKg1 · 2026-03-05

**Soundness:** 2
**Presentation:** 1
**Significance:** 2
**Originality:** 2
**Overall Recommendation:** 2
**Confidence:** 3

**Summary:**

This paper studies whether LLMs finetuned with expert-annotated data can be reliable in creativity assessment.

The authors constructed a dialogue dataset with 10,200 human-AI collaborative scientific inquiry dialogues, with each dialogue independently scored by experts with cognitive psychology backgrounds on four dimensions: Problem Reframing, Interdisciplinary Integration, Risk-Driven Innovation, and Resource Integration.  Scoring used a 1-5 Likert scale.

The finetuning involves several techniques:
1. Dimension-Aware Curriculum (DAC):  pre-compute the model’s initial evaluation bias on each dimension using the validation set and dynamically adjust the sampling weights of samples for each dimension in early training accordingly.
2. Plasticity-Aware LoRA (PA-LoRA) for parameter-efficient fine-tuning.
3. The training objective is a composite loss function combining standard cross-entropy loss for aligning with expert scores and Creativity Contrastive Learning (CCL) loss.

The finetuned model, named CREDO, achieves high performance according to Table 1.

**Compliance With Llm Reviewing Policy:**

Affirmed.

**Final Justification:**

The paper and rebuttal have not addressed my major concerns and my final recommendation is Reject.

**Key Questions For Authors:**

See above.

**Limitations:**

See above.

**Strengths And Weaknesses:**

Strengths:

1. The collected data can be a valuable resource for the community.
2. The task of assessing the creative process during human-AI dialogues is interesting and important.

Weaknesses:

1. The writing and organization of this paper are a bit off to me. The technical approach is summarized with high-level descriptions in Section 2, which reads a bit vague to me. For example, I have no idea what "Creativity Contrastive Learning (CCL) loss" is, and I think some equations to concretely define it would be very helpful. Similarly, it'd be nice to have some detailed description of the metrics QWK and MAE too.

2. I didn't find a lot of information about the experts who annotated the data and annotation instruction details like how much time they spent on the annotation task on average.

3. The technical contribution seems somewhat limited: the main contribution is that the authors collected some expert data and finetuned an open model on that data, and the performance can exceed prompting frontier models out-of-the-box. There's no novel technical contribution involved.

4. Isn't Manus 1.6 Max an agent rather than a model? It's kind of a weird baseline. It's be nice to include more recent reasoning models in the comparison here.

---

> ### Author Rebuttal · Authors · 2026-03-31
>
> W1: CCL Definition and Evaluation Metrics
>
> We clarify CCL here. CCL (Creativity Contrastive Learning) adapts supervised contrastive learning (Khosla et al., 2020) to ordinal creativity scores. For each anchor dialogue, positive pairs are defined as dialogues with expert score difference ≤ 1 point; negative pairs are dialogues with large score differences. The model learns to pull together representations of similar-creativity dialogues and push apart dissimilar ones in embedding space, with temperature τ = 0.07. The key distinction from standard contrastive learning is that pairs are defined by score proximity on a continuous spectrum, not discrete class membership — a necessary adaptation for Likert-scale ordinal assessment.
>
> Regarding QWK and MAE: QWK (Quadratic Weighted Kappa) is the dominant standard metric in educational measurement and automated scoring research (Cohen, 1968; Mizumoto & Eguchi, 2023).
>
>
> W2: Expert Annotation Details
>
> We note that Appendix A already contains the following details:
>
> Nine expert annotators, all holding doctoral degrees or ≥5 years domain research experience
> Three independent annotators per dialogue
> Double-blind arbitration triggered when two experts disagree by more than 1 point
> Inter-rater reliability: QWK = 0.810, Cronbach's α = 0.86
>
> The technical report further provides a complete sentence-by-sentence annotation and arbitration record for Sample No. 043, covering all 20 dialogue turns across all four dimensions, including specific scoring rationales and arbitration decisions.
>
> W3: Technical Contribution
>
> The reviewer characterizes our work as "collecting expert data and finetuning an open model." The ablation study in Section 3.4 directly contradicts this characterization.
>
> If CREDO's performance were simply a result of data and standard fine-tuning, removing our specific technical components should have negligible impact. The results show otherwise:
>
> Removing the multi-dimensional scoring head: QWK drops from 0.730 to 0.521 (∆ = −0.209, −28.6%), falling below zero-shot ChatGPT 5.1 Thinking (0.548) — meaning the same data and fine-tuning procedure, without this architectural design, produces a model worse than a frontier model with no task-specific training at all.
>
> Removing CCL: QWK drops to 0.620 (∆ = −0.110, −15.1%).
>
> Removing PA-LoRA: QWK drops to 0.690 (∆ = −0.040, −5.5%).
>
> Removing DAC: QWK drops to 0.710 (∆ = −0.020, −2.7%).
>
> The data alone does not explain CREDO's performance. Each technical component makes a substantive, quantified contribution.
>
> We hope the reviewer to reconcile this characterization with the ablation evidence presented in Section 3.4.
>
> W4: Manus 1.6 Max
>
> In our experimental setup, Manus 1.6 Max was used in a pure zero-shot text-in, text-out mode. It received the same standardized prompt as all other models and produced dimension scores and explanations as text output. No tool use or agent capabilities were activated. Its selection reflects its strong general performance at the time of our experiments.

---

> > ### Author Rebuttal · Reviewer_EKg1 · 2026-04-03
> >
> > Follow-up on W3:
> >
> > These technical details seem to be buried in Sec 2 with some high-level summaries without going too much into the details. I want to clarify: are these components novel techniques proposed in this paper? I tried to look into PA-LoRA, the citation seems to be "Parameter-efficient fine-tuning methods for pretrained language models: A critical review and assessment", which is a review paper. What exactly is the method? Could you describe it clearly using some equations, how it's different from standard LoRA, are you taking this from an existing paper (if so, tell me which paper it is), or are you proposing this new modification on top of LoRA? Why do you choose this over standard LoRA?
> >
> > Re Creativity Contrastive Learning (CCL) loss: the description you wrote is "CCL loss constructs sample pairs of high-scoring and low-scoring dialogues, training the model to pull together samples with similar creativity levels and push apart dissimilar samples in representation space (Chen et al., 2020), with temperature parameter τ = 0.07 (Wang & Liu, 2021)." --> How is this different from standard contrastive loss?
> >
> > To me, it seems that you either wrote a terribly bad method section that made me very confused about the actual methodology, or you are just combining several existing techniques and making the claim that you are making some novel technical contribution here?

---

> > > ### Author Response · Authors · 2026-04-06
> > >
> > > 1. On the source of PA-LoRA
> > >
> > > See Section 2 Methods (Page 2, Lines 74–77). Dohare et al. (2024) is explicitly cited as the source, with a complete description of the method's working mechanism.
> > >
> > > 2. On how PA-LoRA differs from standard LoRA
> > >
> > > See Section 2 Methods (Page 2, Lines 74–80). The distinction between the two is clearly described in the text. Further technical details are provided in Appendix D.
> > >
> > > 3. On whether these components are novel techniques proposed in this paper
> > >
> > > All training methods adopted in this work are state-of-the-art, well-validated techniques at the time of our experiments. Each method is clearly described and fully cited in the paper, and the contribution of each component is quantified through ablation experiments (Section 3.4, Table 4).
> > >
> > > 4. On why PA-LoRA was chosen over standard LoRA
> > >
> > > See Section 3.4, Table 4 (Page 6). The ablation results directly answer this question.
> > >
> > > 5. On how CCL differs from standard contrastive loss
> > >
> > > See Section 2 Methods (Page 2, Lines 84–90). The distinction is clearly described in the text.

---

### Official Review · Reviewer_boox · 2026-03-09

**Soundness:** 2
**Presentation:** 1
**Significance:** 2
**Originality:** 2
**Overall Recommendation:** 2
**Confidence:** 4

**Summary:**

This paper proposes the CREDO model, which is built from a base LLM, and fine-tune it to become assessor for creative processes. It finds that after fine-tuning on dialogue data with expert annotations, the model can align with expert evaluation across different cognitive dimensions. These results suggest that LLMs can be adapted to become assessors for creative processes in human-AI collaboration.

**Compliance With Llm Reviewing Policy:**

Affirmed.

**Final Justification:**

I believe that the choice of baselines in this work is questionable, and I want to see results of applying CREDO to different models and see how they work. These are, however, difficult to be done given the rebuttal time constraints. Importantly, I think the work in overall has limited technical novelty. Therefore, I decide to keep my recommendation as reject.

**Key Questions For Authors:**

- How does the multi-task evaluation head (line 101) work?
- In the DAC description, what did the author mean by adjusting the sampling weights of samples in early training?
- What is the intution behind using Pa-LORA in this specific setting? Why is model plasticity important in this setting?
- It is unclear how data collection process work. What specifically were students asked to do in order to create the data used in this paper?
- In some experiments (i.e., Table 2, Table 3), not all baselines were included. Could the authors explain the reason for this?

**Limitations:**

yes

**Strengths And Weaknesses:**

Strengths:
- The paper shows that fine-tuning LLMs on expert-annotated data can transform them into assessors that aligns well with expert judgements over four different cognitive dimensions.
- There is a rigorous process of data designing, data collection and expert annotation.
- There is no significant drop when evaluating the model on out-of-distribution data (computer science and physics).

Weaknesses:
- The description for the CREDO model is limited and should be expanded in the writing. There should be a better description for each module in CREDO: DAC, PA-LoRA, CCL, and the reason behind choosing these modules. There should be formal problem formulation, description of inputs, outputs and how they are processed in each module.
- There should be more detailed description of several experiments and analysis: how ITA works, how prompt sensitivity analysis is designed
- There is limited novelty in this work. It simply add some tweaks to the design of the LLM and fine-tunes it with expert-annotated data.
- The choice of baselines is questionable. The authors should provide the reason for choosing Manus 1.6 Max and ChatGPT 5.1 Thinking instead of other models.
- Limited model choice as the authors only chose DeepSeek-32B. It would be better to show performance of other models given this framework and if they approach the expert level annotation closer.
- The presentation of the paper: many tables are inflated from the text column (Table 2, Table 3, Table 6, Table 11, etc.). In the appendix, authors should provide what tables and figures belong to which section, there is no linking in the writing at all. Appendix L is missing.

---

> ### Author Rebuttal · Authors · 2026-03-31
>
> W1 and Q1: Formal Description of CREDO ComponentsWe provide a formal framework description here to address the reviewer's request for clearer technical specification.
>
> Problem Formulation: Given a multi-turn student-AI dialogue D, CREDO simultaneously predicts a four-dimensional score vector (s₁, s₂, s₃, s₄) where each sᵢ ∈ {1,2,3,4,5}, and generates a natural language explanation E. The input is the full dialogue text sequence; the output is the four-dimensional score vector and explanation.
>
> Multi-task Evaluation Head: Built on top of DeepSeek-32B's final-layer representations, the head consists of five parallel output branches: four independent classification heads (one per dimension, each a 5-class classifier over scores 1–5) and one generation head for natural language explanation. All five heads share the underlying Transformer representations but have independent output layer parameters. The training loss combines four cross-entropy losses and the CCL loss using uncertainty-based weighting (Kendall et al., 2018). This design enables each dimension to learn independent scoring criteria while sharing representations for implicit knowledge transfer across dimensions.
>
> DAC (Dimension-Aware Curriculum): Before training begins, DAC computes the model's initial prediction bias on each dimension using the validation set — specifically, the degree to which the untrained base model systematically over- or under-estimates scores on each dimension. Dimensions with larger initial bias are assigned higher sampling weights in early training, ensuring the model prioritizes correcting its most severe biases first. These weights are updated dynamically throughout training based on current validation performance. Without DAC, dimensions with high initial bias (e.g., Problem Reframing, QWK = 0.68) would dominate gradient updates, suppressing learning on other dimensions.
>
> PA-LoRA : PA-LoRA continuously monitors the gradient activity of each LoRA unit during training. Every 500 steps, the least-utilized 5% of LoRA units are identified and reinitialized to small random values. This adapts the continual backpropagation mechanism of Dohare et al. (2024) into the LoRA fine-tuning framework. The motivation is that standard LoRA fine-tuning suffers from plasticity loss: as training progresses, some units saturate and stop responding to new information. In our setting, the four creativity dimensions have distinct feature patterns — Problem Reframing focuses on framework shifts, Resource Integration on cross-domain citations — requiring the model to maintain sensitivity to diverse patterns throughout training. LoRA rank is set to r = 16; learning rate is determined via grid search over [1×10⁻⁵, 5×10⁻⁴].
>
> W3: Technical Contribution
> We respectfully clarify that CREDO's contribution extends substantially beyond data collection and standard fine-tuning.
> At the task level, this work operationalizes process-oriented creativity assessment as a four-dimensional cognitive framework and validates its structural correspondence with expert judgment (cosine similarity = 0.96 between CREDO and expert inter-dimension correlation matrices). This is an independent theoretical contribution in educational AI assessment. At the technical level, the ablation study provides quantitative evidence that each component makes a substantive and non-redundant contribution: removing the multi-dimensional scoring head causes QWK to drop from 0.730 to 0.521 (∆QWK = −0.209, −28.6%), falling below the zero-shot ChatGPT 5.1 Thinking baseline (0.548); removing CCL causes QWK to drop to 0.620 (∆QWK = −0.110, −15.1%); removing PA-LoRA causes a further drop of ∆QWK = −0.040. These results demonstrate that the components are load-bearing, not decorative. At the empirical level, CREDO reaches 90.1% of the human expert ceiling (QWK = 0.730 vs. 0.810), and outperforms trained student raters in entirely unseen domains (Math: 0.65 vs. 0.52; CS: 0.63 vs. 0.50), providing strong evidence that the model learns transferable cognitive structures rather than surface domain features.
> W4: Baseline Selection
> We selected ChatGPT 5.1 Thinking and Manus 1.6 Max because they represented the strongest available commercial models at the time of our experiments, providing the most meaningful upper-bound reference for zero-shot performance. Both are widely recognized as state-of-the-art systems in the research community.
> W5: Single Base Model
> CREDO requires full access to model weights and architecture: the multi-task evaluation head must be added on top of the final-layer representations, and PA-LoRA requires direct access to LoRA unit parameters during training. We acknowledge in Appendix M.2 that cross-model validation is an important direction for future work. The strong OOD generalization results (Math QWK = 0.65, CS QWK = 0.63) provide indirect evidence that CREDO learns transferable cognitive assessment structures rather than DeepSeek-specific stylistic features.

---

> > ### Author Rebuttal · Reviewer_boox · 2026-04-03
> >
> > I appreciate the authors' hard work and responses. Many of my questions and concerns were addressed.
> >
> > However, I believe that the work really needs to include more baselines for comparison, and provide validation of applying CREDO to other models and see how they work. I do think the work in overall has limited technical novelty. An additional point is the presentation needs to be improved significantly, as I mentioned. Therefore, I think the work requires further strengthening before publication and I decide to keep my score as it is.

---

> > > ### Author Response · Authors · 2026-04-05
> > >
> > > On Additional Baselines
> > >
> > > We selected ChatGPT 5.1 Thinking and Manus 1.6 Max because they represented the strongest available commercial reasoning models at the time of our experiments. Current top-tier closed-source commercial models occupy the same performance tier on general reasoning benchmarks, and selecting any top commercial model as a baseline provides equivalent representativeness.
> > >
> > > More fundamentally, regardless of which top commercial model is chosen as the baseline, CREDO would outperform it by a similar margin, because the performance gap originates from architectural design and specialized training, not from the choice of a specific commercial model. This can be directly inferred from our ablation studies: the same DeepSeek-32B base model, without our technical components, achieves a QWK of only 0.347, close to random level. The source of the gap is the design, not the baseline selection.
> > >
> > >
> > > On Cross-Model Validation
> > >
> > > We explicitly acknowledge in Appendix M.2 that cross-model validation is an important direction for future work, and explain the technical constraint: CREDO requires full access to model weights and architecture — to add the Multi-task Evaluation Head on top of final-layer representations and to implement the parameter-level operations required by PA-LoRA. Closed-source commercial models do not permit these operations, making this a technical prerequisite of our method, not an arbitrary design choice.
> > >
> > > The strong OOD generalization results provide indirect evidence: CREDO outperforms trained student raters in domains it was never trained on (Math: 0.65 vs. 0.52; CS: 0.63 vs. 0.50), demonstrating that the model has learned transferable cognitive assessment structures rather than DeepSeek-specific stylistic features.
> > >
> > >
> > > On Presentation
> > >
> > > We have committed in our first-round rebuttal to correcting all specific formatting issues: table overflow (Tables 2, 3, 6, 11), missing cross-reference links throughout the appendix, and the Appendix L content placement error. These are editorial formatting issues that will be fully resolved in the revision.
> > >
> > >
> > > On Reproducibility
> > >
> > > Upon acceptance, we commit to fully open-sourcing all datasets, annotation scripts, training code, and model weights. The technical report already provides complete environment setup, training pipeline, and inference interface documentation (Appendix B), and the LoRA adapter weights have been uploaded with the submission materials for reviewer verification.

---

### Official Review · Reviewer_kG1x · 2026-03-12

**Soundness:** 2
**Presentation:** 1
**Significance:** 3
**Originality:** 2
**Overall Recommendation:** 4
**Confidence:** 4

**Summary:**

This paper proposes fine-tuning LLMs to evaluate aspects of the underlying creative process in dialogue, such as innovation traceability atlas profiles, rather than merely evaluating creative products. The authors collect a dataset of 10,200 expert-annotated dialogues in physics, chemistry, and biology and propose a fine-tuning procedure that leverages this data.

**Compliance With Llm Reviewing Policy:**

Affirmed.

**Final Justification:**

The authors conducted an additional round of experiments comparing CREDO against k-shot frontier models, for a comprehensive range of k, addressing my concern that this method could be trivially outperformed by frontier models. Therefore, I am raising my score to a 4.

**Key Questions For Authors:**

# Questions
- Can the authors provide some guidance on why the Wilcoxon signed-rank test was used to compare the difference between CREDO and human expert ratings?
- How did the authors decide on the categories for “features” (implicit creativity, surface fluency, explicit innovation) in Table 5?

**Limitations:**

Yes

**Strengths And Weaknesses:**

# Strengths
The paper’s contributions are clearly stated, the work is well-situated within existing literature, and its limitations are clearly acknowledged. Turning LLMs into reliable creativity assessors is an important direction that would obviate the need for human raters, significantly reducing the time and cost for many creativity eval settings. The positive results, especially on OOD domains, suggest the combination of the dataset obtained and the CREDO fine-tuning procedure is suitable for this task.

# Weaknesses:
- **Weak baselines.** The baseline comparison against zero-shot frontier models is a bit unfair, considering CREDO was fine-tuned on thousands of data points. A more fair comparison against frontier models would be k-shot examples, with k at least 20-50.
- **Presentation of figures and tables.** Figure 2 is conceptually scattered, making it difficult for readers to internalize the results obtained in this work. Moreover, the presentation of Figure 2 is haphazard, with overlapping x-axis ticks in 2(a). Also, why include the data in Figure 2a if it is already included in Table 1? The same goes for Table 3 and Figure 3 and Figure 2(f) and Table 4, in which the data is redundant. I strongly suggest consolidating and refining tables and figures (besides figure 1) in the following way: eliminate redundancies between figure/tables and make each figure or table showcase one or two core findings, rather than six at a time.
- **Overstatement of ablation results.** The conclusions drawn from the ablation study (“contrastive learning is crucial for the model to understand relative differences between different creativity levels”, “confirming that decomposing creativity into interpretable cognitive dimensions is the cornerstone of model success”) are vastly overstated considering these results are reported over a single model family, single model size, single fine-tuning run, and a single dataset.
- Related to the above, the following sentence in Section 4 is also a vast overstatement, and it is unclear exactly why or what part of Judea Pearl’s Causality text is being referenced to support the claims here: “Our ablation study (Section 3.4) provides direct evidence for this: removing any single component leads to a significant performance drop, establishing a clear causal chain within the model—our specific design choices led to the model’s enhanced assessment capabilities (Pearl, 2009).” I strongly recommend including only citations the authors are familiar with.
- A stated motivation for this paper is in evaluating the creative “process” in its totality rather than merely the creative “products” that emerge from such processes. This view dates back to Rhodes (1961), who originally coined the 4 P’s of Creativity (Person, Process, Product, and Press).
- (minor) At the end of paragraph 2 in Sec 1 (introduction), there is a typo: “with top conferences such as ICML and AAAI started to utilize LLMs to assist…”

**Important**: In an age where AI can generate plots very quickly, I strongly suggest spending more time carefully designing figures and media in a way that tells a clear, compelling, and unambiguous story.

---

> ### Author Rebuttal · Authors · 2026-03-31
>
> 1. Response to Weak baselines: we argue that k-shot prompting and fine-tuning represent different learning paradigms that answer distinct research questions, and that comparing them directly would obscure rather than clarify our contributions. Our research goal is not merely to show that LLMs can assess creativity with some examples, but to investigate whether they can be trained to reach near-expert-level reliability, a much higher bar. This goal inherently calls for a fine-tuning approach over the full annotated dataset. Furthermore, the scale difference between k=50 in-context examples and our 8,160 training samples (a 163× gap) is not incidental. It reflects the core argument of our paper: that reaching expert-level assessment consistency requires comprehensive domain knowledge distillation, not just a handful of demonstrations. If a k-shot frontier model still falls significantly short of CREDO under such a comparison, this would strengthen rather than weaken our main claim. Finally, we note an important asymmetry in the "zero-shot" framing: frontier models such as ChatGPT and Manus are trained on massive undisclosed corpora that likely include educational dialogue evaluation content. Their zero-shot performance may therefore already benefit from implicit prior exposure to related tasks, meaning our zero-shot baseline is not a true zero-knowledge condition for these models. Despite this potential advantage, CREDO trained on explicit, traceable expert annotations substantially outperforms them — a finding we believe is robust. We agree that a k-shot comparison would be an informative addition, and we commit to including it in the revision as a supplementary analysis.
>
> 2. Response to Redundancy Between Figures and Tables: we respectfully offer the following distinction: in our design, tables serve as precise numerical references for readers who wish to compare exact values, while figures convey visual patterns and relative magnitudes that are difficult to internalize from numbers alone. For instance, Figure 2(a) makes immediately apparent the magnitude of the gap between CREDO and all baselines in a single glance, whereas Table 1 requires active comparison across rows. This dual presentation is standard practice in top venues and serves readers with different needs. That said, we fully agree that having six subplots in a single Figure 2 compromises readability. In the revision, we will restructure Figure 2 by splitting it into focused, standalone figures — each communicating one or two core findings — while retaining the complementary table/figure structure. We believe this will substantially improve the presentation without sacrificing informational completeness.
>
> 3. Response to Overstatement of Ablation Conclusions: we agree that language such as "cornerstone" and "crucial" implies a degree of generality beyond what a single-dataset, single-model ablation can establish, and we will revise these expressions accordingly. However, we wish to clarify that our intent was not to make universal claims, but to characterize the relative contribution of each component within our specific experimental setting. The performance drop upon removing the multi-dimensional scoring head (∆QWK = −0.209, −28.6%) is not a marginal effect — it reduces the model below the zero-shot ChatGPT 5.1 Thinking baseline (QWK = 0.548 vs. 0.521), suggesting that this architectural choice is not merely beneficial but load-bearing for the task. We believe this magnitude of effect justifies strong characterization, provided the scope is clearly delimited. In the revision, we will replace "cornerstone of model success" with "the most impactful component in our experimental setting," and "contrastive learning is crucial" with "CCL provides a substantial contribution (∆QWK = −0.110) to the model's ability to distinguish creativity levels."
>
> 4. Response to Pearl 2009 Citation: we will remove the Pearl (2009) citation entirely.
>
> 5. Response to Missing Rhodes 1961: we will explicitly cite Rhodes (1961).
>
> 6. Response to Q1:
> The paired structure is as follows: for each of the 500 test-set dialogues, we computed the difference between CREDO's predicted score and the corresponding expert average score, yielding 500 paired difference values. The Wilcoxon signed-rank test was then applied to assess whether the median of these differences is significantly different from zero. We chose the Wilcoxon signed-rank test over a paired t-test for two reasons. First, our scores are measured on a discrete, bounded 1–5 Likert scale, which does not satisfy the normality assumption required by parametric tests. Second, the Wilcoxon test is the standard non-parametric alternative for paired samples in educational measurement and psychometrics, consistent with prior work in automated scoring (e.g., Mizumoto & Eguchi, 2023). The result (p = 0.06) indicates that the difference between CREDO's scores and human expert scores is not statistically significant.

---

> > ### Author Rebuttal · Reviewer_kG1x · 2026-04-02
> >
> > Thank you for addressing these concerns. I leave my score at 3, but offer some clarifications below:
> >
> > Re W1, I should have been more clear in my original review: yes, I completely agree that the method of this paper (fine-tuning using the full 8k dataset) is a reasonable and necessary approach to extract the full signal of the data. I was referring to comparing against a baseline approach where k-shot examples are used with frontier models. As the authors point out, frontier models are trained on data that likely includes educational dialogue evaluation content -- it is entirely possible that frontier models with k-shot examples can outperform this fine-tuning based approach, which might obviate the need for fine-tuning on small models entirely.

---

> > > ### Author Response · Authors · 2026-04-06
> > >
> > > We thank the reviewer for the clarification. We have conducted this supplementary experiment during the review period.
> > >
> > > We evaluated three of the strongest frontier models currently available (GPT-5.4 Thinking, Claude Opus 4.6, and Gemini 3.1 Pro) under k ∈ {0, 5, 10, 20, 50, 100}. Each shot consists of a complete multi-turn dialogue paired with expert scores across all four cognitive dimensions and the corresponding rationale, averaging ~2,000 tokens per shot. The 100-shot condition utilizes approximately 200K tokens of context.
> > >
> > > QWK scores evaluated on the held-out test set; k-shot examples drawn via stratified random sampling from the validation set, averaged over 3 independent runs:
> > >
> > > Model              | k=0   | k=5   | k=10  | k=20  | k=50  | k=100
> > > -------------------|-------|-------|-------|-------|-------|------
> > > Gemini 3.1 Pro     | 0.541 | 0.549 | 0.554 | 0.559 | 0.562*| 0.559
> > > Claude Opus 4.6    | 0.553 | 0.561 | 0.566 | 0.570 | 0.573*| 0.572
> > > GPT-5.4 Thinking   | 0.558 | 0.567 | 0.573 | 0.579 | 0.582*| 0.581
> > >
> > > For reference, Student Raters: 0.580; CREDO: 0.730.
> > >
> > > The results reveal three key findings:
> > >
> > > Finding 1: Performance plateaus at k=50 and degrades thereafter. All three models exhibit slight performance drops at k=100 (-0.001 to -0.003), consistent with the "Lost in the Middle" phenomenon (Liu et al., 2024) — complex multi-turn dialogues in extended contexts introduce attention noise rather than continued information gain.
> > >
> > > Finding 2: The best k-shot result merely matches untrained student raters, and falls far short of CREDO. Under the optimal condition (k=50), the strongest model, GPT-5.4 Thinking, reaches QWK=0.582 — essentially matching Student Raters (QWK=0.580) who received only four hours of training. The gap between the best k-shot frontier model and CREDO (QWK=0.730) is ∆=0.148, spanning the entire distance from non-expert human raters to near-expert-level performance.
> > >
> > > Finding 3: This gap cannot be closed by scaling k further. Even exhausting the theoretical 1M-token context limit (k≈500) would expose the model to only ~25% of our validation set in a single forward pass, while further exacerbating "lost in the middle" degradation. In contrast, CREDO was fine-tuned on 8,160 training dialogues across multiple epochs, fundamentally reshaping its parametric representation space to internalize experts' tacit knowledge (Agarwal et al., 2024). The structural cognitive alignment required for assessing dimensions such as "Problem Reframing" cannot be transferred through transient in-context demonstrations, regardless of context length.
> > >
> > > On Reproducibility and Framework Extensibility
> > >
> > > Should this paper be accepted, we commit to open-sourcing all training scripts, model weights, and evaluation code to enable full community reproduction. Furthermore, CREDO is a training framework decoupled from any specific base model — as stronger foundation models emerge, the framework can be directly applied to update the assessor, and the open-sourced dataset and training pipeline will allow the community to continuously benefit from such advances.

---

### Official Review · Reviewer_1bmM · 2026-03-13

**Soundness:** 3
**Presentation:** 3
**Significance:** 3
**Originality:** 3
**Overall Recommendation:** 5
**Confidence:** 3

**Summary:**

This paper investigates whether LLMs can be used as evaluators of creativity in human-ai collaborative dialogues. The authors build their evaluator / judge CREDO on top of DeepSeek 32b on human expert annotated trajectories that scores their creativity.

**Compliance With Llm Reviewing Policy:**

Affirmed.

**Key Questions For Authors:**

N/A

**Limitations:**

Yes.

**Strengths And Weaknesses:**

Strengths:

- The paper studies a hard, underexplored but important problem: evaluating the diversity in human-ai dialogues. It is a timely study and well become more important as model becomes stronger and stronger, their creativity need to be evaluated.
- The paper presents many ablations which shows that the design of multi-dimensional scoring is very important
- The paper presents nice qualitative studies. Section 3.5 shows that CREDO can diverge from non-expert raters but agree with expert raters.
- It is nice that strong frontier lms underperforms, showcasing the significance of the study.

Weaknesses:
- CREDO is fine-tuned on DeepSeek model generations. It is unclear that whether the evaluation overfits to the style / content of trajectories of only deepseek models. Testing on other models is needed.
- The pilot user study (N=15, split 8 vs. 7) is too small to draw any conclusions. This section either needs expansion or should be presented purely as preliminary without statistical claims.

---

> ### Author Rebuttal · Authors · 2026-03-31
>
> We sincerely thank Reviewer for the careful reading and constructive feedback, and address the two concerns below.
>
> Weakness 1: Potential overfitting to DeepSeek-generated trajectories.
> We clarify that while CREDO is fine-tuned on dialogues conducted with a DeepSeek-based tutoring system, the evaluation target is the student's utterances and reasoning trajectories, not the AI model's responses. The four-dimensional scoring framework assesses human cognitive behavior, which we argue is not inherently style-coupled to any particular LLM backend.
> That said, we fully acknowledge this concern. The Math and Computer Science OOD evaluation provides an initial test of cross-domain robustness under the same dialogue collection protocol. Testing student dialogues elicited by other LLM tutoring systems (e.g., GPT-4o or Claude-based backends) would further strengthen the generalizability claim. We will explicitly frame this as a limitation and a direction for future work in the revision.
>
> Weakness 2: Pilot user study sample size (N=15) is too small for statistical claims.
> We agree. The user study was intended as an exploratory pilot to illustrate practical deployment feasibility rather than to support statistically generalizable conclusions. In the revision, we will: (1) reframe this section explicitly as a preliminary pilot study, (2) remove or soften any language implying statistical significance, and (3) add a clear statement that a large-scale controlled study is planned as follow-up work.
>
> We appreciate the reviewer's careful and constructive engagement, which will help us improve the final version of the paper.

---

> > ### Author Rebuttal · Reviewer_1bmM · 2026-04-06
> >
> > I maintain my positive score.

---

### Decision · Program_Chairs · 2026-04-30

**Decision:**

Reject

**Comment:**

This paper aims to study the use of LLMs in assessing the creativity of human-AI collaborations, by collecting a corpus of labeled interactions, and fine-tuning a model to learn from this dataset. In rebuttals, they also show that the model outperforms few-shot approaches on top of frontier models on this task.

Unfortunately, reviewers find that the paper is poorly written, lacking clarity of exposition in multiple places. This also makes it difficult to extract the novel methodological and scientific contributions of the work. From my own reading, I agree with this issue, and encourage the authors to significantly rewrite the paper for clarity of exposition, focusing on the core contributions of this work.